# Metabolic response of blood vessels to TNFα

**Abidemi Junaid[1,2,3], Johannes Schoeman[1], Wei Yang[1], Wendy Stam[2,3], Alireza Mashaghi[1], Anton Jan van Zonneveld[2,3], Thomas Hankemeier[1]***

[1]Division of Systems Biomedicine and Pharmacology, Leiden Academic Centre for Drug Research, Leiden University, Leiden, Netherlands; [2]Department of Internal Medicine (Nephrology), Leiden University Medical Center, Leiden, Netherlands; [3]Einthoven Laboratory for Vascular and Regenerative Medicine, Leiden University Medical Center, Leiden, Netherlands

**Abstract** TNFα signaling in the vascular endothelium elicits multiple inflammatory responses that drive vascular destabilization and leakage. Bioactive lipids are main drivers of these processes. In vitro mechanistic studies of bioactive lipids have been largely based on two-dimensional endothelial cell cultures that, due to lack of laminar flow and the growth of the cells on non-compliant stiff substrates, often display a pro-inflammatory phenotype. This complicates the assessment of inflammatory processes. Three-dimensional microvessels-on-a-chip models provide a unique opportunity to generate endothelial microvessels in a more physiological environment. Using an optimized targeted liquid chromatography–tandem mass spectrometry measurements of a panel of pro- and anti-inflammatory bioactive lipids, we measure the profile changes upon administration of TNFα. We demonstrate that bioactive lipid profiles can be readily detected from three-dimensional microvessels-on-a-chip and display a more dynamic, less inflammatory response to TNFα, that resembles more the human situation, compared to classical two-dimensional endothelial cell cultures.

*For correspondence:
hankemeier@lacdr.leidenuniv.nl

## Introduction

Tumor necrosis factor-α (TNFα) is a central mediator of the inflammatory response (*Sedger and McDermott, 2014*). TNFα can be generated by monocytes or macrophages and activates endothelial cells at sites of tissue injury or infection through TNF receptor-1 (TNFR1) (*Chimen et al., 2017*; *Torres-Castro et al., 2016*; *Green et al., 2016*). Following activation, the endothelium elicits a multitude of local responses such as vascular leakage, leukocyte adhesion and coagulation that together are essential to the physiological homeostatic responses to anti-microbial immunity.

However, chronic exposure to adverse metabolic and hemostatic risk factors (*Masi et al., 2018*), obesity (*Engin, 2017*), or disease states such as kidney disease (*Rabelink et al., 2010*) or rheumatoid arthritis (*van Zonneveld et al., 2010*) are all associated with a systemic inflammatory condition and elevated circulating levels of TNFα. As a consequence, TNFα signaling induces the generation of high levels of free radicals in the vascular endothelium that, when excessive, can deplete the cellular anti-oxidant defense systems and lead to a state of oxidative stress and vascular dysfunction (*Pisoschi and Pop, 2015*).

Mechanistically, TNFα signaling in endothelial cells involves the activation of NFκB and results in the increased synthesis of reactive oxygen species (ROS) from a number of sources such as mitochondria, NADPH oxidase, uncoupled eNOS, xanthine oxidase, and peroxidases (*Cai and Harrison, 2000*; *Blaser et al., 2016*). On its turn, elevated ROS can lead to the generation of bioactive lipids directly or indirectly, such as prostaglandins, isoprostanes, lysophosphatidic acid classes, sphingolipids and platelet activating factor (PAF). Under physiologic conditions, in concert with the

**eLife digest** In a range of conditions called autoimmune diseases, the immune system attacks the body rather than foreign elements. This can cause inflammation that is harmful for many organs. In particular, immune cells can produce excessive amounts of a chemical messenger called tumor necrosis factor alpha (TNFα for short), which can lead to the release of fatty molecules that damage blood vessels.

This process is normally studied in blood vessels cells that are grown on a dish, without any blood movement. However, in this rigid 2D environment, the cells become 'stressed' and show higher levels of inflammation than in the body. This makes it difficult to assess the exact role that TNFα plays in disease.

A new technology is addressing this issue by enabling scientist to culture blood vessels cells in dishes coated with gelatin. This allows the cells to organize themselves in 3D, creating tiny blood vessels in which fluids can flow. However, it was unclear whether these 'microvessels-on-a-chip' were better models to study the role of TNFα compared to cells grown on a plate.

Here, Junaid et al. compared the levels of inflammation in blood vessels cells grown in the two environments, showing that cells are less inflamed when they are cultured in 3D. In addition, when the artificial 3D-blood vessels were exposed to TNFα, they responded more like real blood vessels than the 2D models. Finally, experiments showed that it was possible to monitor the release of fatty molecules in this environment. Together, this work suggests that microvessels-on-a-chip are better models to study how TNFα harms blood vessels.

Next, systems and protocols could be develop to allow automated mass drug testing in microvessels-on-a-chip. This would help scientists to quickly screen thousands of drugs and find candidates that can protect blood vessels from TNFα.

transcriptional regulation of a plethora of inflammatory genes (*Poussin et al., 2020*), these bioactive lipids are critically involved in the first response of endothelial cells to environmental changes, controlling vascular permeability and platelet- and leukocyte adhesion. Also, in fore mentioned patients, isoprostanoids such as 8-epiPGF2α are generated by peroxidation during conditions of oxidative stress (*Morrow et al., 1990*) and serve as gold standard oxidative stress plasma markers (*Ridker, 2004*) associating with an increased risk for cardiovascular disease and its underlying causes (*Moutzouri et al., 2013*; *Vassalle et al., 2004*).

The central role of TNFα in many disease states has identified this cytokine as an important therapeutic target to counteract vascular inflammation (*Sedger and McDermott, 2014*; *Esposito and Cuzzocrea, 2009*) and several TNFα blockers have been approved by the FDA and have been effective for the suppression of immune-system diseases, such as Crohn's disease, ulcerative colitis, rheumatoid arthritis, ankylosing spondylitis, psoriatic arthritis and plaque psoriasis (*Ackerman et al., 2016*).

Following the success of the TNFα antagonists, a deeper understanding of the impact of oxidative stress on the homeostasis of the human vascular endothelium may yield more specific targets for inflammatory diseases affecting the vasculature. To that end, many in vitro mechanistic studies have relied on static, two-dimensional (2D) cultures of primary endothelial cells such as those derived from human umbilical veins (HUVECs) (*Jaffe et al., 1973*). However, in recent years it has become increasingly apparent that these cultures reflect a 'stressed' endothelial phenotype due to the lack of their native environmental cues. In vitro, endothelial cells are usually cultured on surfaces such as plastics and glass that are much stiffer than natural substrates such as the extracellular matrix. Recent studies demonstrated that primary endothelial cells on a hard substrate adopt a pro-inflammatory phenotype (*Stroka and Aranda-Espinoza, 2011*; *Huveneers et al., 2015*). Vascular stiffness is strongly associated with vascular disorders such as arterial hypertension, kidney disease and atherosclerosis (*Huveneers et al., 2015*). Likewise, native microvessels need laminar shear to maintain a quiescent phenotype and the lack of laminar shear stress in static cultures converts endothelial cells to a pro-inflammatory 'diseased' phenotype (*Baeyens et al., 2016*).

Novel microfluidics-based perfused three-dimensional (3D) microvessels-on-a-chip models provide a unique opportunity to generate endothelial microvessels in a more physiological environment.

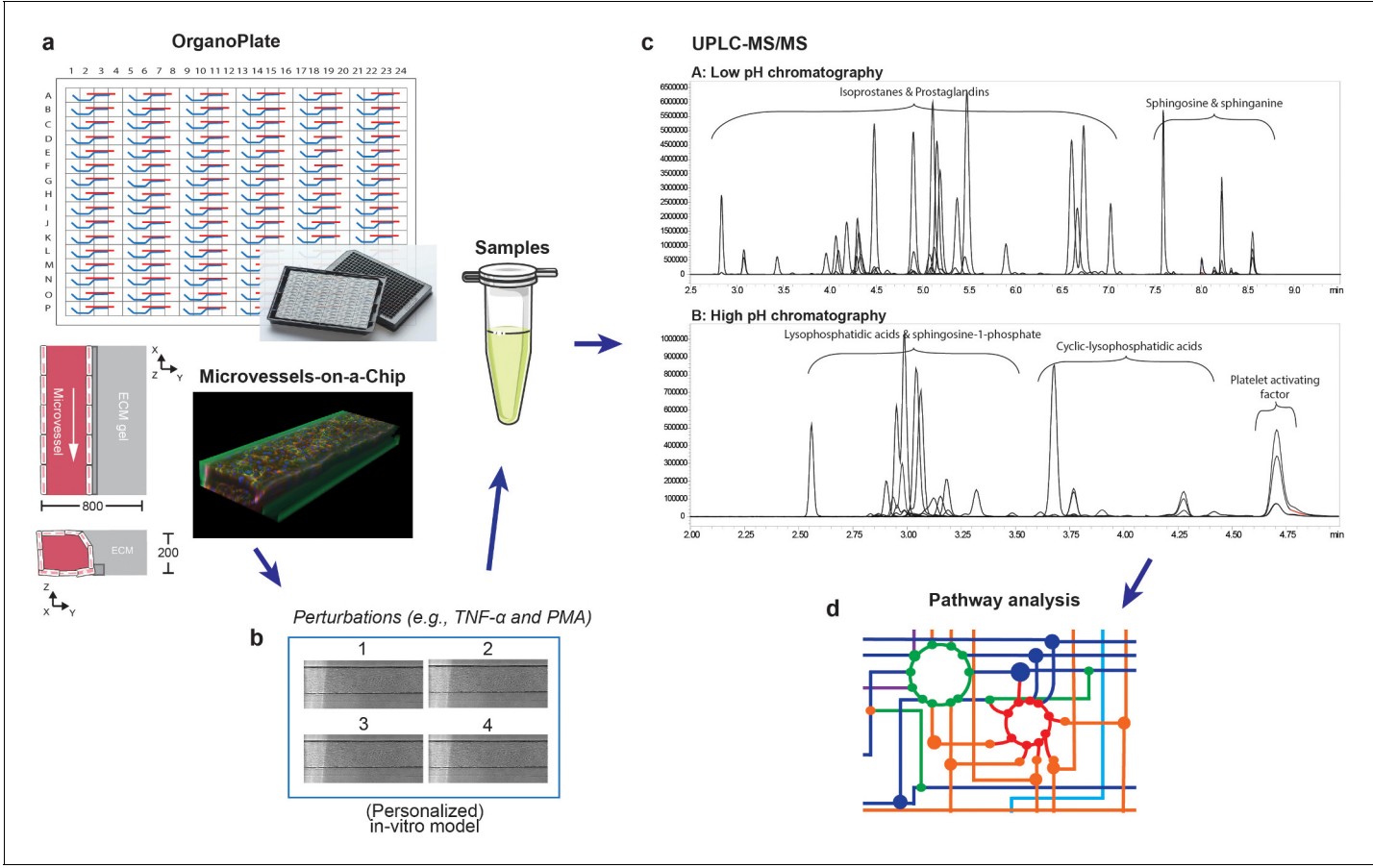

**Figure 1.** Metabolomics workflow. (**a**) Schematic diagram of the OrganoPlate 2-lane design and 3D reconstruction of the microvessels-on-a-chip formed by cultured HUVECs (blue: Hoechst, red: F-actin and green: VE-cadherin). All dimensions are in μm. (**b**) Collection of culture media after perfusion. The medium of four microvessels were pooled to form one sample. (**c**) Identification and quantification of prostaglandins, isoprostanes, lysophosphatidic acid (LPA) classes, sphingolipids and platelet activating factor (PAF) in microvessels-on-a-chip by UPLC-MS/MS using two different solvent gradients. (**d**) Pathway analysis.

We employed a gelatin coated 3D microvessels-on-a-chip model in which endothelial cells are organized in a tube-like architecture, with densities of cells and area to volume ratios that are closer to a physiological condition than those in typical 2D culture.

To assess whether these 3D microvessels display a more anti-inflammatory phenotype, we used an optimized targeted liquid chromatography–tandem mass spectrometry measurements of a panel of pro- and anti-inflammatory bioactive lipids and generated expression profiles both in TNFα treated microvessels under flow as well as in 2D endothelial cell cultures under static condition. We demonstrate bioactive lipid profiles can be readily detected from single microvessels and display a more dynamic, less inflammatory response to TNFα, that resembles more the human situation, compared to classical 2D endothelial cell cultures.

## Results and discussion

### Bioactive lipids generated by 3D microvessels-on-a-chip can be measured by UPLC-MS/MS

In this paper, we present a novel set-up to measure the metabolic response of 3D endothelial microvessels to TNFα, including pro- and anti-inflammatory markers. We cultured 96 perfused microvessels against extracellular matrix (ECM) using the microvessels-on-a-chip platform technology recently developed by using the OrganoPlate platform of MIMETAS (*Wevers et al., 2018*; *van Duinen et al., 2017*). The microchannels in the OrganoPlate were coated with gelatin, preventing endothelial cells

**Table 1.** The peak area ratio of metabolites in the culture medium (EGM2) normalized with the peak area ratio of metabolites found in the culture medium after perfusion in the microvessels-on-a-chip for 18 hr (EGM2 HUVECs).

The peak area ratio is the peak area of the metabolites divided by the appropriate peak area of the internal standards. Fold changes below the 1 (blue) and above the 1 (red) indicates that low and high concentrations of fatty acids were present in medium before exposure to the microvessels. The data represent one biological replicate; n = 3 technical replicates.

| Bioactive lipid* | EGM2/EGM2 HUVECs | Bioactive lipid* | EGM2/EGM2 HUVECs |
|---|---|---|---|
| PGF2α | 0.1 | LPA C22:5 | 18.2 |
| PGF3α | 2.1 | LPA C16:0 | 21.3 |
| 8-iso-13, 14-dihydro-PGF2α | 0.0 | LPA C18:1 | 48.4 |
| 8-iso-PGF2α | 0.2 | LPA C22:4 | 5.9 |
| 5-iPF2α | 0.4 | cLPA C20:4 | 78.6 |
| 8, 12-iPF2α IV | 0.5 | LPA C18:0 | 0.0 |
| LPA C14:0 | 6.2 | cLPA C18:2 | 0.0 |
| LPA C16:1 | 25.4 | cLPA C16:0 | 14.8 |
| LPA C22:6 | 17.7 | cLPA C18:1 | 25.8 |
| LPA C18:2 | 77.2 | cLPA C18:0 | 11.1 |
| LPA C20:4 | 31.0 | S-1-P C18:1 | 0.9 |

* The rest of the metabolites shown in *Figure 3* are not displayed, because they were not detected in the EGM2.

The online version of this article includes the following source data for Table 1:

Source data 1. Peak area ratios of the identified metabolites in culture medium (EGM2) and in culture medium after perfusion in the microvessels-on-a-chip (EGM2 HUVECs).

from growing on glass and enabling them to form stable microvesseels. The shear stress in the microvessels, calculated based on a previous work, ranges from 1 to 5 dyne/cm$^2$ (*van Duinen et al., 2017*). In vivo, the shear stress ranges from 95.5 dyne/cm$^2$ at the smallest capillaries to 2.8 dyne/cm$^2$ at the postcapillary venules (*Koutsiaris et al., 2007*). Conditioned medium perfused through TNFα treated and control (untreated) microvessels was sampled, pooled and measured with a UPLC-MS/MS metabolomics method developed recently by us to study inflammation and oxidative stress (*Figure 1*; *Schoeman et al., 2018*).

As a metabolic read-out for TNFα signaling, we measured prostaglandins, isoprostanes, LPAs, lysosphingolipids and PAF and first assessed the concentrations of these metabolites in the basic EGM2 medium and compared their concentrations after 18 hr incubation to condition the medium in the microvessels-on-a-chip cultures. As shown in *Table 1*, the peaks detected demonstrated that, while members of the prostaglandins and isoprostanes clearly increased after conditioning of the medium, most of the LPA metabolites were readily detectable in the medium and actually displayed a significant decreased concentration. When the EGM2 medium was incubated for 18 hr without

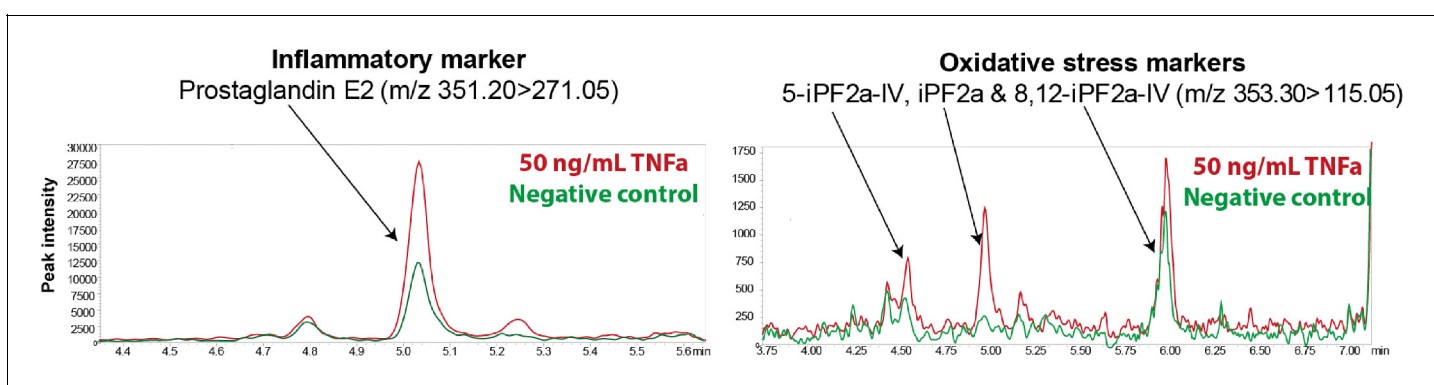

**Figure 2.** Inflammatory and oxidative stress markers in microvessels-on-a-chip. Reconstructed LC-MS/MS ion chromatograms of PGE2, 5-iPF2α IV, iPF2α and 8, 12-iPF2α IV in microvessels treated with 50 ng/ml TNFα for 18 hr.

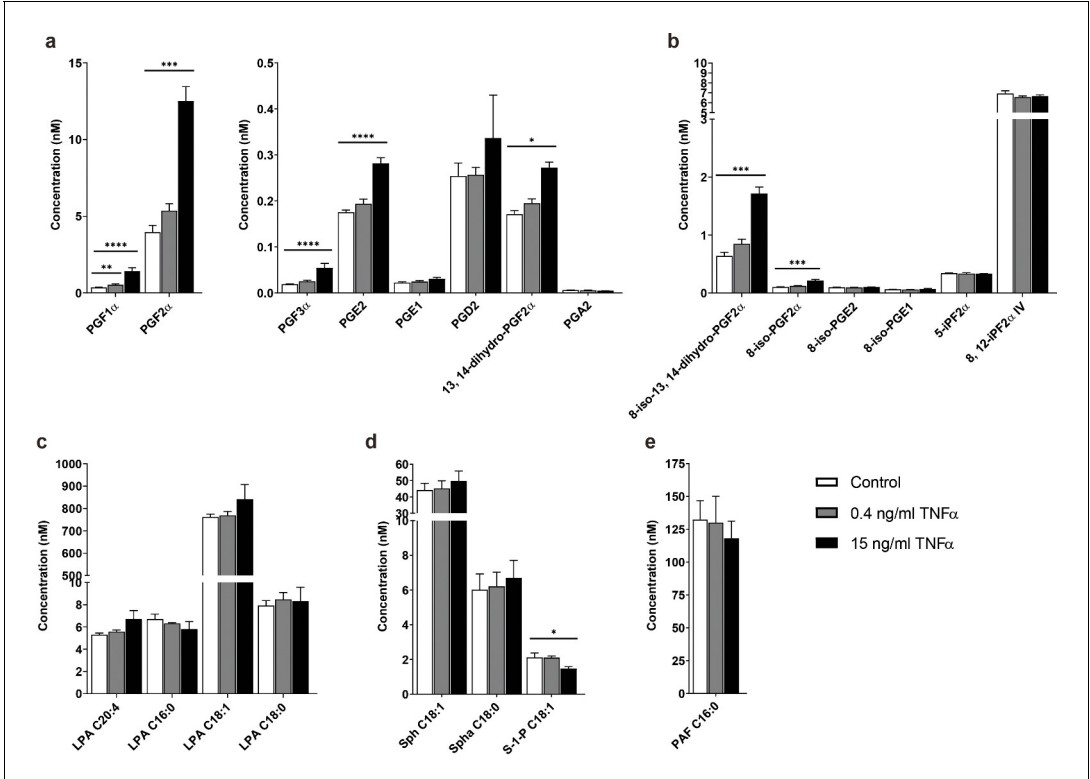

**Figure 3.** TNFα-induced concentration profile changes of the signaling lipids in the microvessels-on-a-chip. Concentrations of (**a**) prostaglandins, (**b**) isoprostanes, (**c**) lysophosphatidic acid (LPA) classes, (**d**) sphingolipids and (**e**) platelet activating factor (PAF) with available standards detected in the microvessels without TNFα exposure (control) and after exposure to 0.4 and 15 ng/ml TNFα for 18 hr. Data represent mean and s.e.m. of three biological replicates per condition; n = 4–6 technical replicates. Significance determined by unpaired Student's t-test; *p<0.1, **p<0.05, ***p<0.01, ****p<0.001.

exposure to the microvessels no significant changes in the concentrations of these metabolites was observed. Therefore, we conclude that during the conditioning of the medium, prostaglandins and isoprostanoids are excreted from the endothelial cells and that the LPA metabolites are consumed or actively degraded by the cells.

When we assessed the biolipid composition of the conditioned medium of TNFα stimulated microvessels, 33 measured metabolites passed the quality control (QC) thresholds. *Figure 2* shows examples of the chromatograms of prostaglandin E2 and different isoprostanes isomers that were secreted by the microvessels after exposure to TNFα for 18 hr compared to the control samples showing a marked increase in abundance of a number of these metabolites. The control samples are microvessels without exposure to TNFα. Bar plots of the absolute concentrations of prostaglandins are presented in *Figure 3a*, showing a significant difference between untreated and TNFα treated microvessels. While at physiological concentration of TNFα (0.4 ng/ml) there was an increase in the excretion of PGF1α, PGF2α, PGF3α, PGE2, PGD2 and 13, 14-dihydro-PGF2α, however, no significant difference between the untreated and TNFα treated microvessels was evident, except for PGF1α. At 15 ng/ml a stronger differential response was observed for selected isoprostanes, several LPAs, sphingolipids and PAF (*Figure 3b–e*). The relative concentrations of the bioactive lipids found in the microvessels-on-a-chip are strikingly similar with those found in normal human blood vessels (*Table 2*).

Phorbol 12-myristate 13-acetate (PMA) is an activator of protein kinase C (PKC), hence of NFκB, that relates to TNFα signaling and causes a wide range of effects in cells. As it is a known and potent up-regulator of cyclooxygenase-2 (COX-2) (*Chang et al., 2005*), we next measured the effect of PMA on the secretion of the biolipids to compare the TNFα response to a condition of maximal stimulation. When our combined results were plotted in a heatmap, marked differences are observed between control- and TNFα or PMA-treated microvessels (*Table 3*).

**Table 2.** Comparison of the concentration of bioactive lipids between living human blood vessel and human microvessels-on-a- chip.
The concentrations in human blood vessel were obtained from HMDB (*Wishart et al., 2018*; *Wishart et al., 2013*; *Wishart et al., 2007*; *Wishart et al., 2009*).

| Bioactive lipid | Human blood vessel | | Microvessels-on-a-chip | |
|---|---|---|---|---|
| | Healthy | Diseased | Healthy | Diseased |
| PGF1α | ~0.0317–0.376 nM | - | ~0.350 nM | ~0.527–1.412 nM |
| PGF2α | ~0.144–0.371 nM | ~0.4–1.6 nM | ~3.96 nM | ~5.36–12.5 nM |
| PGE2* | ~0.13–0.172 nM | - | ~0.175 nM | ~0.194–0.281 nM |
| PGE1 | <0.1 nM | - | ~0.0225 nM | ~0.0246–0.0308 nM |
| PGD2 | ~0.065–0.2 nM | - | ~0.254 nM | ~0.257–0.336 nM |
| PGA2 | ~0.0448–0.496 nM | - | ~0.006 nM | ~0.0048–0.0058 nM |
| 8-iso-PGF2α | ~0.057–0.57 nM | - | ~0.103 nM | ~0.122–0.216 nM |
| S-1-P C18:1 | ~0.5–3.0 nM | - | ~2.12 nM | ~1.47–2.11 nM |
| Sph C18:1 | ~1.3–50 nM | - | ~44.2 nM | ~45.2–49.8 nM |
| Spha C18:0 | ~1.3–50 nM | - | ~6.0 nM | ~6.2–6.7 nM |

## Impact of TNFα on prostaglandin levels

As shown in *Table 3*, overnight exposure to TNFα and PMA shows increase in the release of the prostaglandins PGF1α, PGF2α, PGF3α, PGE2, PGD2 and 13, 14-dihydro-PGF2α from the TNFα treated microvessels (*Figure 3a* and *Table 3*). During inflammation, ROS contributes to the increased PGE2, PGF2α, PGD2 and 13, 14-dihydro-PGF2α production through the release of arachidonic acid and COX-2 activation, having a pro-inflammatory effect in the endothelium (*Wong and Vanhoutte, 2010*; *Dworski et al., 2001*). At the same time, anti-inflammatory prostaglandins PGF1α, PGF3α, PGE1, and PGA2 are also secreted by the endothelium (*Trebatická et al., 2017*; *Gezginci-Oktayoglu et al., 2016*). PGE1 and PGA2 are known to suppress TNFα induced NFκB activation and production of ROS (*Ohmura et al., 2017*). Relating to these two prostaglandins, no differences were detected in our system between untreated and TNFα treated microvessels for 18 hr.

## Impact of TNFα on isoprostane levels

When we focus on the compounds produced by the reaction of free radicals with arachidonic acid, the isoprostanes, high levels of 8-iso-13, 14-dihydro-PGF2α and 8-iso-PGF2α were detected in the supernatant of TNFα treated microvessels (*Figure 3b* and *Table 3*). These metabolites inhibit platelet aggregation and induce monocyte adhesion to endothelial cells (*Rokach et al., 1997*; *Durack-ová, 2010*). We also detected 8-iso-PGE2, 5-iPF2α, 8, 12-iPF2α IV and 8-iso-PGE1 in the control sample and TNFα induced microvessels. 8-iso-PGE1 is recognized as vasoconstrictor with a similar effect as PGF2α (*Nakano and Kessinger, 1970*). However, no significant difference between the two groups was evident after incubating the microvessels for 18 hr with TNFα.

## Impact of TNFα on lysophosphatidic acids, sphingolipids and platelet activating factor

Looking at lipids that mediate diverse biological actions, the LPA classes, sphingosine and PAF are appropriate markers to take along in our metabolic read-out, because of their diverse biological actions. The LPA classes consist of LPAs and cyclic-lysophosphatidic acids (cLPAs). They are formed by activated platelets and oxidation of low-density lipoproteins (LDLs) (*Karshovska et al., 2018*). Once an inflammatory response is triggered, LPAs can activate platelets (*Khandoga et al., 2008*) and lead to endothelial dysfunction by activating NFκB (*Biermann et al., 2012*; *Yang et al., 2016*; *Ninou et al., 2018*; *Palmetshofer et al., 1999*). On the other hand, cLPAs inhibit pro-inflammatory cytokine expression in the endothelium (*Tsukahara et al., 2014*). In our data, we saw high concentrations of several LPAs in the control sample compared to TNFα treated microvessels. Similar results were seen in the levels of

**Table 3.** Heatmap of prostaglandins, isoprostanes, lysophosphatidic acid (LPA) classes, sphingolipids and platelet activating factor (PAF) detected in the microvessels-on-a-chip.

The fold changes were measured with respect to the controls and log2 transformed. The controls are microvessels unexposed to TNFα and PMA. The metabolites are characterized by their inflammatory action (anti- or pro-inflammatory), platelet activation (anti- or pro-platelet activation), vascular tone (constriction or dilation) and angiogenic action (anti- or pro-angiogenic). The data were obtained from the experiments done in **Figure 3** with three biological replicates per condition; n = 4–6 technical replicates.

| | Bioactive lipid | Fold change of concentration | | | Inflammatory | Platelet | Vascular | Angiogenic |
|---|---|---|---|---|---|---|---|---|
| | | 15 ng/ml | 50 ng/ml | 20 ng/ml | | | | |
| | | TNF | TNF | PMA | action | activation | tone | action |
| Prostaglandins | PGF1α | 2.0 | 1.8 | 5.0 | anti | no | con | |
| | PGF2α | 1.7 | 1.5 | 5.0 | pro | no | con | pro |
| | PGF3α | 1.5 | 1.1 | 4.4 | anti | | | |
| | PGE2* | 0.7 | 0.7 | .7 | pro | anti | dil | pro |
| | PGE1 | 0.5 | 0.4 | 2.7 | anti | anti | dil | pro |
| | PGD2 | 0.4 | 3.4 | 3.5 | anti | anti | con | anti |
| | 13, 14-dihydro-PGF2α | 0.7 | 0.5 | 2.3 | pro | | | |
| | PGA2 | −0.3 | 0.0 | 2.6 | anti | no | | |
| Isoprostanes | 8-iso-13, 14-dihydro-PGF2α | 1.4 | 1.3 | 4.6 | | | | anti |
| | 8-iso-PGF2α* | 1.1 | 0.9 | 4.2 | pro | anti | con | anti |
| | 8-iso-PGE2 | 0.1 | 0.0 | 2.1 | pro | anti | con | anti |
| | 8-iso-PGE1 | 0.1 | 0.0 | 0.7 | | anti | con | anti |
| | 5-iPF2α | 0.0 | 0.0 | 0.0 | | | | |
| | 8, 12-iPF2α IV | −0.1 | 0.0 | 0.2 | | | | |
| Lysophosphatidic acids | LPA C14:0 | −0.2 | −0.2 | −0.4 | pro | pro | con | pro |
| | LPA C16:1 | −0.4 | −0.3 | −0.6 | pro | pro | con | pro |
| | LPA C22:6* | 0.4 | 0.5 | 0.2 | pro | pro | con | pro |
| | LPA C18:2 | 0.1 | 0.0 | −0.1 | pro | pro | con | pro |
| | LPA C20:4 | 0.3 | 0.4 | 0.3 | pro | pro | con | pro |
| | LPA C22:5* | 0.5 | 0.6 | 0.3 | pro | pro | con | pro |
| | LPA C16:0 | −0.2 | −0.3 | −0.3 | pro | pro | con | pro |
| | LPA C18:1 | 0.1 | 0.2 | −0.1 | pro | pro | con | pro |
| | cLPA C20:4 | −0.1 | −0.2 | −0.1 | anti | anti | no | |
| | LPA C18:0 | 0.1 | 0.0 | −0.2 | pro | pro | con | pro |
| | cLPA C16:0 | −0.2 | 0.0 | 0.0 | anti | anti | no | |
| | cLPA C18:0 | −0.2 | −0.1 | −0.2 | anti | anti | no | |
| Sphingolipids | S-1-P C18:1 | −0.5 | −0.6 | −0.9 | anti | anti | con | pro |
| | Sph C18:1 | 0.2 | 0.1 | 0.0 | anti | anti | con | pro |
| | Spha C18:0 | 0.2 | 0.0 | −0.1 | | | | |
| | PAF C16:0 | −0.2 | −0.2 | −0.4 | pro | pro | con | pro |

* Validated markers of oxidative stress.

The online version of this article includes the following source data for Table 3:

**Source data 1.** Concentrations of the identified metabolites in the microvessels-on-a-chip.

sphingosine-1-phosphate (S-1-P) and PAF (*Table 3* and *Figure 3c–e*). In TNFα signaling, S-1-P binds to TNF receptor-associated factor 2 (TRAF2) to activate NFκB, while PAF induces vascular permeability (*Alvarez et al., 2010*; *Palur Ramakrishnan et al., 2017*).

**Table 4.** Heatmap of pro- and anti- inflammatory and oxidative stress markers measured in 3D microvessels-on-a-chip and 2D endothelial cell monolayers.

The cells were treated with 15 ng/ml TNFα in the same experiment as *Figure 3*. The fold changes were measured with respect to the controls and log2 transformed. The controls are microvessels unexposed to TNFα and PMA. The metabolites are characterized by their inflammatory action (anti- or pro-inflammatory), platelet activation (anti- or pro- platelet activation), vascular tone (constriction or dilation) and angiogenic action (anti- or pro-angiogenic). The data represent one biological replicate; n = 2–3 technical replicates.

| Bioactive lipid | Fold change of Concentration | | Inflammatory | Platelet | Vascular | Angiogenic |
| | 2D | 3D | action | activation | tone | action |
| | TNF | TNF | | | | |
| --- | --- | --- | --- | --- | --- | --- |
| PGF1α | 1.9 | 3.4 | anti | no | con | |
| PGF3α | 1.0 | 6.6 | anti | | | |
| PGE1 | 2.1 | 1.9 | anti | anti | dil | pro |
| PGD2 | 2.3 | 6.5 | anti | anti | con | anti |
| PGA2 | 0.7 | 0.0 | anti | no | | |
| cLPA C20:4 | −0.7 | −0.5 | anti | anti | no | |
| cLPA C18:2 | −0.5 | 0.0 | anti | anti | no | |
| cLPA C16:0 | −0.6 | −0.3 | anti | anti | no | |
| cLPA C18:1 | −0.9 | −0.2 | anti | anti | no | |
| cLPA C18:0 | −0.6 | −0.2 | anti | anti | no | |
| S-1-P C18:1 | −2.0 | −0.9 | anti | anti | con | pro |
| 8-iso-PGE1 | 1.8 | 1.9 | | anti | con | anti |
| 5-iPF2α | 0.3 | −0.1 | | | | |
| PGF2α | 1.9 | 2.2 | pro | no | con | pro |
| PGE2* | 2.4 | 1.0 | pro | anti | dil | pro |
| 13, 14-dihydro-PGF2α | 0.9 | 1.1 | pro | | | |
| 8-iso-13, 14-dihydro-PGF2α | 1.9 | 1.9 | | | | anti |
| 8-iso-PGF2α* | 2.0 | 1.3 | pro | anti | con | anti |
| 8-iso-PGE2 | 0.6 | −0.3 | pro | anti | con | anti |
| LPA C14:0 | 0.0 | −0.8 | pro | pro | con | pro |
| LPA C16:1 | −1.0 | −1.0 | pro | pro | con | pro |
| LPA C22:6* | 0.0 | −0.3 | pro | pro | con | pro |
| LPA C18:2 | −1.0 | −1.2 | pro | pro | con | pro |
| LPA C20:4 | −0.2 | −0.4 | pro | pro | con | pro |
| LPA C22:5* | 1.0 | −0.1 | pro | pro | con | pro |
| LPA C16:0 | −0.6 | −0.6 | pro | pro | con | pro |
| LPA C18:1 | −0.5 | −1.0 | pro | pro | con | pro |
| LPA C18:0 | 0.1 | −0.9 | pro | pro | con | pro |
| PAF C16:0 | −0.5 | −0.8 | pro | pro | con | pro |

* Validated markers of oxidative stress.

The online version of this article includes the following source data for Table 4:

Source data 1. Peak area ratios of the identified metabolites in 6-well plates and in the microvessels-on-a-chip.

## TNFα induced bioactive lipid profiles from endothelial cells in 3D configuration are less inflammatory compared to 2D monolayers

To assess whether these three-dimensional microvessels display a more anti-inflammatory phenotype, we compared the bioactive lipid response of the 3D microvessels-on-a-chip to TNFα to that of 2D endothelial cell monolayers. In addition, we made a detailed inventory of the reported action of

the individual lipids on inflammation, platelet activation, vascular tone and angiogenesis (for references see *Supplementary file 1*). When the TNFα-induced biolipids profiles are listed in relation to their biological activities (*Table 4*), we conclude that the 3D microvessels-on-a-chip display a more dynamic, less inflammatory response to TNFα, that resembles more the human situation, compared to classical 2D endothelial cell cultures. In particular, the anti-inflammatory prostaglandins PGF1α, PGF3α, and PGD2 are increased to a larger extent and the anti-inflammatory lysophosphatidic acids are maintained or decreased to a lesser extent. In concert, the pro-inflammatory lipids PGE2, 8-iso-PGE2 and 8-iso-PGF2α are present at higher levels in the medium of the TNFα exposed 2D endothelial monolayer culture. The elevated levels of the oxidative stress markers PGE2, 8-iso-PGF2α (*Ridker, 2004*), LPA C22:5 and LPA C22:6 (*Ackerman et al., 2016*) confirm the increased inflammatory status of the 2D cultures making it tempting to speculate that an increased production of ROS in these cells may underlie these responses. The response of the microvasculature to inflammatory cytokines such as TNFα is often directly associated with inhibition of platelet activation (to maintain patency of the microvessel), a vasoconstrictive response and a pro-angiogenic response characterized by the loss of endothelial cell-cell contacts and microvascular leakage. Many of these activities are also driven by the bioactive lipids in our panel (*Supplementary file 1*), and it is interesting to note that concomitantly to the less inflammatory nature of the profiles of the conditioned media derived from the 3D microvessel the profile also suggests to be more restrictive of platelet activation, less vasoconstrictive and less angiogenic. It should be noted that we did not compare the excretion of bioactive of unstimulated 2D cell cultures and 3D vessels as the normalization on the number of endothelial cells would not have been straightforward.

The observed differences between the 2D and the 3D chip-based platforms may be attributed to the mechanical properties of the two systems (*Lee et al., 2009*). The microvessels-on-a-chip are surrounded by an ECM layer and the 3D configuration allows intensified cell-cell interactions, resembling the in vivo situation. Moreover, vascular endothelial cells in vivo are influenced by distinct hemodynamic forces and this applies also to the endothelial cells in our microvessels-on-a-chip. Evidence suggest that shear stress activates phospholipids turnover that is involved in the production of free arachidonic acid (*Bhagyalakshmi et al., 1992*). This might also explain the differences we see between the increase/decrease of fatty acids in the microvessels-on-a-chip and the 2D cell culture. As shear stress influences RhoA activity and stress fiber formation, the regulation of fatty acids, RhoA might be important in this process (*Liu et al., 2014*). In addition, the environmental changes in the 3D configuration could impact on the expression of the TNF receptors.

Several reports showed that oxidative stress induces endothelial dysfunction, which plays a central role in vascular diseases. It can promote the expression of pro-inflammatory and pro-coagulant factors, apoptosis and impair the release of nitric oxide (*Schulz et al., 2011*; *Shiraki et al., 2012*). This study set out with the aim of using metabolomics as a readout of endothelial function in microvessels-on-a-chip exposed to TNFα, to trigger inflammatory responses seen in vasculopathy. For the first time we show that the regulation of prostaglandins, isoprostanes, LPAs, sphingolipids and PAF can be measured in our microfluidic system, even though they cause profound physiological effects at very dilute concentrations that serve as early-stage markers of oxidative stress and inflammation (*Ryu et al., 2015*). The findings support the model that TNFα signaling induces ROS production that causes changes in signal transduction and gene expression, which leads to release of oxidative stress and inflammatory markers (*Figure 4*). Further research should be undertaken to confirm the results in gene and protein levels.

## Conclusions

We demonstrate bioactive lipid profiles can be readily detected from minor volumes of <1 μl of conditioned medium from microvessels-on-a-chip and display a more dynamic, less inflammatory response to TNFα compared to classical two-dimensional endothelial cell cultures. We can conclude that the response to TNFα resembles for the microvessels-on-a-chip more the human situation as described in the literature than the 2D endothelial cell culture. As the physiological readout of endothelial function is a critical aspect in using microvessels-on-a-chip for disease and drug research, the results suggest that the metabolic readout using metabolomics is more informative compared to morphological changes studied with imaging analyses of phenotypic changes. But it is the combination of both techniques, metabolic readout using metabolomics and imaging analysis that may facilitate mechanistic studies and the detection and validation of biomarkers for microvascular disease at

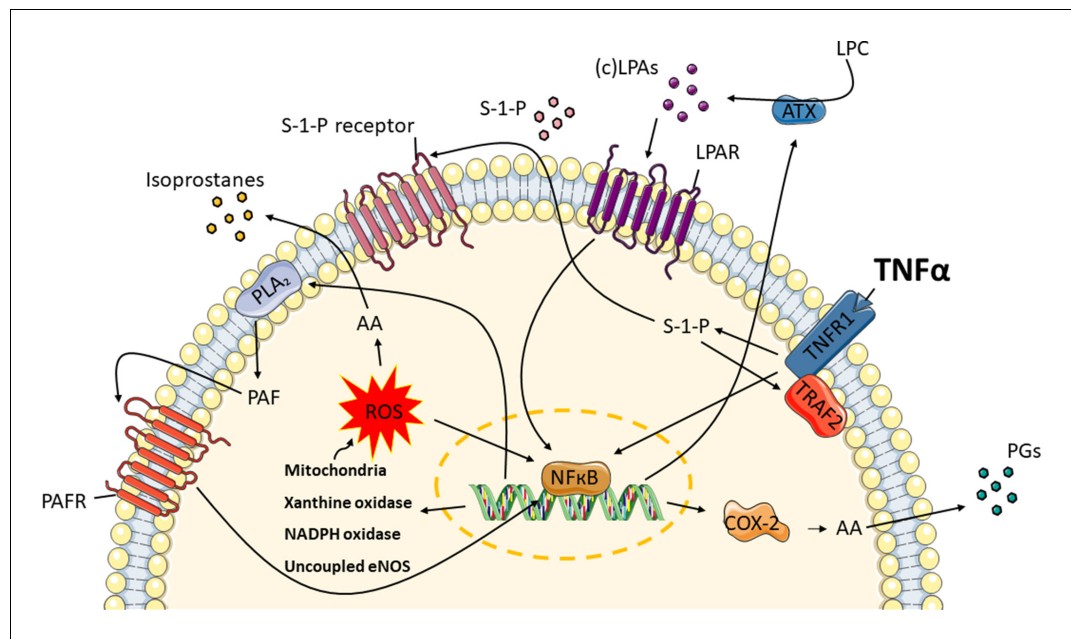

**Figure 4.** TNFα induces the release of oxidative stress and inflammatory markers in endothelial cells. Exposure to TNFα, causes TNF signaling in the microvessels to produce ROS from endogenous sources: mitochondria, xanthine oxidase, NADPH oxidase and uncoupled eNOS. Sphingosine-1-phosphate (S-1-P) is needed in order for TNF receptor-associated factor 2 (TRAF2) to form a complex with the TNF receptor 1 (TNFR1). These lead to the conversion of arachidonic acid (AA) to isoprostanes and NFκB activation. Moreover, AA is enzymatically converted by cyclooxygenase-2 (COX-2) to prostaglandins (PGs). At the same time, autotaxin (ATX) and phospholipase A$_2$ (PLA$_2$) are upregulated, resulting in the syntheses of lysophosphatidic acid (LPA) classes and platelet activating factor (PAF). Through their receptors, LPAs and PAF further promote the activation of COX-2.

the systemic level. Furthermore, it will provide the information needed to understand microvascular destabilization and will generate a knowledge base for developing and testing personalized therapeutic interventions.

## Materials and methods

**Key resources table**

| Reagent type (species) or resource | Designation | Source or reference | Identifiers | Additional information |
|---|---|---|---|---|
| Biological sample (Human) | primary human umbilical vein endothelial cells | Leiden University Medical Center (LUMC) | | freshly isolated from umbilical cord of male newborns |
| Chemical compound, drug | phorbol 12-myristate 13-acetate | Sigma-Aldrich | Cat#:P8139 | |
| Peptide, recombinant protein | tumor necrosis factor-α | Sigma-Aldrich | Cat#:H8916 | |
| Biological sample (Rat) | rat tail collagen type 1 | Trevigen | Cat#:3440-005-01 | |
| Antibody | mouse anti-human CD144 | BD Biosciences | Cat#:555661; RRID:AB_396015 | IF(1:150) |
| Antibody | sheep anti-human CD31 | R and D Systems | Cat#:AF806; RRID:AB_355617 | IF(1:150) |
| Antibody | rabbit anti-human vWF | Agilent Dako | Cat#:A0082; RRID:AB_2315602 | IF(1:1000) |

*Continued on next page*

*Continued*

| Reagent type (species) or resource | Designation | Source or reference | Identifiers | Additional information |
|---|---|---|---|---|
| Antibody | alexa fluor 488-conjugated goat anti-mouse | ThermoFisher | Cat#:R37120; RRID:AB_2556548 | IF(1:250) |
| Antibody | alexa fluor 488-conjugated donkey anti-sheep | ThermoFisher | Cat#:A11015; RRID:AB_141362 | IF(1:250) |
| Antibody | alexa fluor 647-conjugated goat anti-rabbit | ThermoFisher | Cat#:A27040; RRID:AB_2536101 | IF(1:250) |
| Other | rhodamine phalloidin | Sigma-Aldrich | Cat#:P1951; RRID:AB_2315148 | IF(1:200) |
| Other | hoechst | Invitrogen | Cat#:H3569; RRID:AB_2651133 | IF(1:2000) |
| Software, algorithm | LabSolutions | Shimadzu | RRID:SCR_018241 | |
| Software, algorithm | SPSS | SPSS | RRID:SCR_002865 | |
| Software, algorithm | GraphPad Prism | GraphPad | RRID:SCR_002798 | |

## Cell culture

Human umbilical vein endothelial cells (HUVECs) were isolated from umbilical cord of newborns, collected with informed consent, by an adaption of the method developed by *Jaffe et al., 1973*. Although denoted as veins, umbilical veins carry oxygenated blood and thus the phenotype of their endothelium is similar to arterial endothelial cells.

The umbilical cord was severed from the placenta soon after birth and placed in a sterile container filled with phosphate-buffered saline (PBS; Fresenius Kabi, The Netherlands) and held at 4°C until processing. The cord was inspected and at both ends a piece of 1 cm was cut off to remove damaged tissue from clamping. Subsequently, the umbilical vein was cannulated and perfused with PBS to wash out the blood and allowed to drain. When clear fluid flow was observed, the vein was filled with trypsin/EDTA solution (CC-5012, Lonza, USA), placed in the container filled with PBS and incubated at 37°C for 20 min. After incubation, the trypsin-EDTA solution containing the endothelial cells was flushed from the cord with air and afterwards PBS. The effluent was collected in a sterile 50 ml tube containing 20 ml Endothelial Cell Growth Medium 2 (EGM2; C-39216, PromoCell, Germany) supplemented with antibiotics and the cell suspension was centrifuged at 1200 rpm for 7 min. The cell pellet was resuspended in 10 ml EGM2 and cultured on 1% gelatin-coated T75 flasks. Cells were maintained in a 37°C incubator with 5% $CO_2$ and the medium was refreshed every other day. After 80% confluency, cells were spilt at 1:3 ratio and cultured in new 1% gelatin-coated T75 flasks. The isolated cells were positive for the endothelial cell markers, including platelet endothelial cell adhesion molecule (PECAM-1) and von Willebrand factor (vWF) (*Figure 5*). All experiments using HUVECs were repeated six times using cells from three different male donors at passage 3.

For 2D experiments, we cultured 50 ·$10^3$ cells/ml in 24-well plates overnight at 37°C in humidified air containing 5% $CO_2$. The following day, the cells were incubated with 15 and 50 ng/ml TNFα (H8916, Sigma-Aldrich, The Netherlands) and 20 ng/ml phorbol 12-myristate 13-acetate (PMA; P8139, Sigma-Aldrich, The Netherlands) for 18 hr. The medium was collected and stored in −80°C.

We used the OrganoPlate (9603-400-B, MIMETAS, The Netherlands) for all microfluidic cell culture experiments. The microvascular and extracellular matrix (ECM) channels were separated by phaseguides (*Vulto et al., 2011*). Before seeding the cells, 4 mg/ml rat tail collagen type 1 (3440-005-01, Trevigen, USA) neutralized with 10% 37 g/L $Na_2CO_3$ (S5761, Sigma-Aldrich, The Netherlands) and 10% 1 M HEPES buffer (15630-056, Gibco, The Netherlands) was added in the ECM channels. Subsequently, the collagen was let to polymerize by incubating the device for 10 min in the incubator at 37°C and 5% $CO_2$. The observation windows were filled with 50 µl Hank's Balanced Salt Solution with calcium and magnesium buffers (HBSS+; 24020117, Life Technologies, The Netherlands) for optical clarity and to prevent gel dehydration. Using a repeater pipette, 2 µl of 1% gelatin was added into the inlet of each microvascular channel and the device was put in the incubator at 37°C

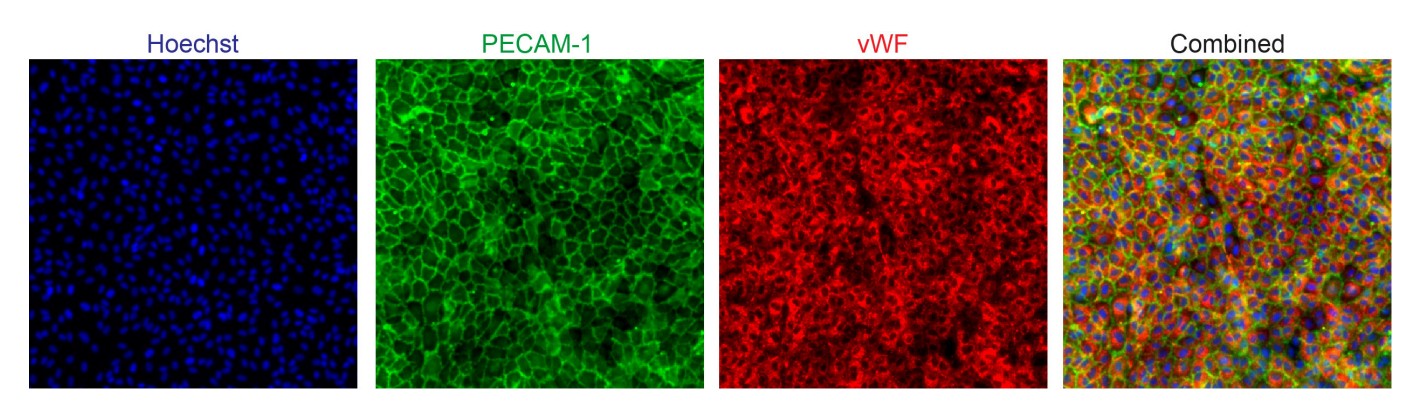

**Figure 5.** Expression of platelet endothelial cell adhesion molecule (PECAM-1) and von Willebrand factor (vWF) in isolated human umbilical vein endothelial cells (HUVECs).

for 30 min. We trypsinized cells at 80-90% confluency and seeded $15 \cdot 10^6$ cells/ml in the outlet of the microvascular channels of the OrganoPlate. Afterwards, the cells were incubated at 37°C and 5% $CO_2$ for one hour to allow microvascular formation. After incubation, 50 µl of culture medium was added to the inlets and outlets of the microvascular channels. The device was placed on a rocker platform with a 7° angle of motion and an eight-minute timed operation to allow continuous flow of minor volumes of medium in the microvessels. The microvascular channels typically contain volumes of <1 µl. After 24 hours, the medium was refreshed, and the HUVECs were cultured for an additional 3-4 days. The microvessels were treated with TNFα (0.4, 15 and 50 ng/ml) and PMA (20 ng/ml) for 18 hours. Subsequently, medium of four microvessels were pooled to form one sample to allow analyses of metabolites at low concentrations due to low cell numbers. This still allowed us to create three biological replicates with 4 – 6 technical replicates data per experimental condition for metabolomics analyses. The samples were stored in -80°C.

## Immunofluorescence staining

For immunofluorescence staining, HUVECs were fixed using 4% paraformaldehyde (PFA) in HBSS+ for 10 min at room temperature. The fixative was aspirated, and the cells were rinsed once with HBSS+. Next, the cells were permeabilized for 2 min with 0.2% Triton X-100 in HBSS+ and washed once with HBSS+. The cells were blocked in 5% BSA in HBSS+ for 30 min and incubated with the primary antibody solution overnight at 4°C. Mouse anti-human CD144 (1:150; 555661, BD Biosciences, USA), sheep anti-human CD31 (1:150; AF806, R and D Systems, The Netherlands) and rabbit anti-human vWF (1:1000; A0082, Agilent Dako, USA) were used as the primary antibodies. The cells were washed with HBSS+, followed by an one-hour incubation with Hoechst (1:2000; H3569, Invitrogen, USA), rhodamine phalloidin (1:200; P1951, Sigma-Aldrich, The Netherlands) and the secondary antibody solution, containing Alexa Fluor 488-conjugated goat anti-mouse (1:250; R37120, Thermo-Fisher, USA), Alexa Fluor 488-conjugated donkey anti-sheep (1:250; A11015, ThermoFisher, USA) and Alexa Fluor 647-conjugated goat anti-rabbit (1:250; A27040, ThermoFisher, USA) antibodies. The cells were washed three times with HBSS+. High-quality Z-stack images of the stained cells were acquired using a high-content confocal microscope (Molecular Devices, ImageXpress Micro Confocal).

## Metabolic profiling

All samples were measured using an oxidative and nitrosative stress profiling platform which has been developed and validated in our lab (*Schoeman et al., 2018*). This platform covers various isoprostane classes, signaling lipids from the sphingosine and sphinganine classes and their phosphorylated forms, as well as three classes of lysophosphatidic acids: lysophosphatidic acids, alkyl-lysophosphatidic acids and cyclic-lysophosphatidic acids (all ranging from C14 to C22 chain length species). For metabolite extraction, sample preparation procedure was according to the in-house experimental protocol which has been standardized and published; extra samples were pooled for

internal quality control (QC) (*Schoeman et al., 2018*). Briefly, cell media (150 µl) were thawed on ice and added with 5 µl antioxidant solution and 10 µl internal standards (ISTDs). Acidified with citric acid/phosphate buffer (pH 4.5), all samples were then dealt with liquid-liquid extraction (LLE) with 1 mL of butanol and ethyl acetate (1:1 v/v). Samples were vortexed and centrifuged and then the organic phase was collected and dried. After reconstitution with ice-cold 70% MeOH injection solution, each sample was again vortexed and centrifuged and the supernatant was transferred to the insert in a glass vial. Ultra-performance liquid chromatography tandem mass spectrometry (UPLC-MS/MS) based analysis was then applied for low-pH measurement (Shimadzu LCMS-8060, Japan) and high-pH measurement (Shimadzu LCMS-8050, Japan) respectively.

## Calibration curve preparation

Standard stock solutions were prepared in MeOH containing butylated hydroxytoluene (0.4 mg/ml). A calibration stock was made with concentrations found in *Supplementary file 2* for the prostaglandins, isopostranes, LPAs, sphingolipids and PAF available base standards and was labeled 'C9'. This solution was diluted to levels C8 to C1 and from these mixes, 20 µl was added to 150 ul sample to construct the calibration curves.

## Data pre-processing

LabSolutions (Shimadzu, Version 5.91) was applied to accomplish all the peak determination and integration. For each metabolite, the response ratio was obtained by calculating the ratio of peak area of the target compound to the peak area of the assigned internal standard. After QC evaluation, metabolites of which QC samples had an RSD less than 30% were used for further statistical analysis. Finally, the absolute concentration of the targets was determined using the calibration curves.

## Statistical analysis

Heatmaps and bar plots were created with GraphPad Prism 7 (GraphPad Software). The fold change was calculated by normalizing the conditions to the control group. Subsequently, the data were log2 transformed and used for the heatmaps. The absolute concentrations of those compounds were visualized in the bar plots. We used IBM SPSS Statistics 23 (IBM) for statistical analyses. Bar plots were plotted as mean ± s.e.m. of three biological replicates per condition; n = 4–6 technical replicates. Significance levels were set at $*p<0.1$, $**p<0.05$, $***p<0.01$, $****p<0.001$ using the unpaired Student's t-test.

# Acknowledgements

This study was financially supported by the RECONNECT CVON Groot consortium, which is funded by the Dutch Heart Foundation. AJ and TH, AJvZ were supported by a ZonMW MKMD grant (114022501). AM and TH acknowledge the support by the NWO-TTW (IMMUNMET, grant number 16249). TH acknowledge the support by the TKI METABOCHIP project, which is co-funded by the PPP Allowance made available by Health~Holland, Top Sector Life Sciences & Health, to stimulate public-private partnerships.

# Additional information

## Competing interests

Thomas Hankemeier: co-founder of MIMETAS and has some shares in MIMETAS. The other authors declare that no competing interests exist.

## Funding

| Funder | Grant reference number | Author |
|---|---|---|
| Hartstichting | RECONNECT CVON Groot | Abidemi Junaid<br>Anton Jan van Zonneveld<br>Thomas Hankemeier |

| ZonMw | 114022501 | Abidemi Junaid<br>Anton Jan van Zonneveld<br>Thomas Hankemeier |
| Nederlandse Organisatie voor Wetenschappelijk Onderzoek | 16249 | Alireza Mashaghi<br>Thomas Hankemeier |
| Health~Holland | TKI METABOCHIP project | Thomas Hankemeier |

The funders had no role in study design, data collection and interpretation, or the decision to submit the work for publication.

## Author contributions

Abidemi Junaid, Conceptualization, Investigation, Methodology, Writing - original draft, Writing - review and editing; Johannes Schoeman, Investigation, Methodology, Writing - review and editing; Wei Yang, Investigation, Writing - review and editing; Wendy Stam, Investigation; Alireza Mashaghi, Supervision, Writing - review and editing; Anton Jan van Zonneveld, Thomas Hankemeier, Conceptualization, Resources, Supervision, Funding acquisition, Writing - review and editing

## Author ORCIDs

Abidemi Junaid (iD) https://orcid.org/0000-0001-8562-7942
Johannes Schoeman (iD) http://orcid.org/0000-0003-0905-2467
Wei Yang (iD) http://orcid.org/0000-0002-3394-7570
Thomas Hankemeier (iD) https://orcid.org/0000-0001-7871-2073

## Decision letter and Author response

Decision letter https://doi.org/10.7554/eLife.54754.sa1
Author response https://doi.org/10.7554/eLife.54754.sa2

## Additional files

### Supplementary files

• Source data 1. Calibration curve of bioactive lipids.

• Supplementary file 1. References regarding the action of bioactive lipids on inflammation, platelets, vascular tone and angiogenesis.

• Supplementary file 2. An overview of the concentrations of the calibration solution.

• Transparent reporting form

### Data availability

The data used to generate the figures and tables can be found in the source data files.

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
