## [Decision Letter]

**Acceptance summary:**

The study integrates two new and innovative technologies [use of 1) a 3-D cell culture model and 2) metabolites across different lipid classes] as an in vitro model to investigate the metabolic effects of TNF-α on blood vessels. The results of the study provide important new tools and insights into blood vessels and how they operate under different inflammatory burden.

**Decision letter after peer review:**

Thank you for submitting your article "Metabolic response of blood vessels to TNFα" for consideration by *eLife*. Your article has been reviewed by two peer reviewers, and the evaluation has been overseen by a Reviewing Editor and Matthias Barton as the Senior Editor. The following individual involved in reviewing your submission has agreed to reveal their identity: Warwick Dunn (Reviewer #2).

The reviewers have discussed the reviews with one another and the Reviewing Editor has drafted this decision to help you prepare a revised submission.

Summary:

The study integrates two innovative technologies [use of 1) a 3-D cell model and 2) metabolites across different lipid classes] to investigate the metabolic effects of TNF-α on blood vessels. The results of the study provide important new tools and insights into blood vessels and how they operate under different inflammatory burden.

Essential revisions:

– Absolute concentration of the lipids measured should be shown. The log 2 ratios in the comparisons in tables are frequently negative, which may suggest that detected amounts are very low. The use of a microchip with flow in low volumes may be a challenge to detect metabolites. It would also be important to compare the absolute levels detected in the in vitro model to what is observed in vivo.

– The number of replicates analysed appears to be mostly 2 after pooling of samples; this provides a limitation of the manuscript. According to the authors, "Bar plots were plotted as mean ± SEM of two or three technical replicates. Significance levels were set at *p < 0.05 using the Mann-Whitney U test." In Supplementary Figure 1, however, the n number given is 4. The authors used only n=2 in most of the experiments after pooling from several samples. According to the authors the "Mann-Whitney U test" was used to calculate statistical comparisons, a test that applies to data with non-parametric distribution. Distribution of the data indicated by the error bars in many experiments suggests parametric distribution. An n=2 of replicates does neither allows to determine data distribution nor normality/non-normality, let alone to choose the appropriate statistical test. In view of this and given the fact that individual replicates were pooled from more samples it is recommended that the authors consult a biostatistician regarding how to best compare and analyse these data and add this new information to the manuscript. Alternatively, either the replicates for each experiment would need to be increased to an n number sufficient to determine distribution normality and to apply the corresponding test, or any information indicating statistical significances would need to be removed.

---

## [Author Response]

Essential revisions:– Absolute concentration of the lipids measured should be shown. The log 2 ratios in the comparisons in tables are frequently negative, which may suggest that detected amounts are very low. The use of a microchip with flow in low volumes may be a challenge to detect metabolites. It would also be important to compare the absolute levels detected in the in vitro model to what is observed in vivo.

We have now included the absolute concentration of the lipids. This is shown in Figure 3 and an explanation regarding the calibration curve is given in the Materials and methods section. Furthermore, we compared the absolute concentrations of some of the detected targets in our microvessels-on-a-chip with what is observed in vivo. This is shown in Table 2.

– The number of replicates analysed appears to be mostly 2 after pooling of samples; this provides a limitation of the manuscript. According to the authors, "Bar plots were plotted as mean ± SEM of two or three technical replicates. Significance levels were set at *p < 0.05 using the Mann-Whitney U test." In Supplementary Figure 1, however, the n number given is 4. The authors used only n=2 in most of the experiments after pooling from several samples. According to the authors the "Mann-Whitney U test" was used to calculate statistical comparisons, a test that applies to data with non-parametric distribution. Distribution of the data indicated by the error bars in many experiments suggests parametric distribution. An n=2 of replicates does neither allows to determine data distribution nor normality/non-normality, let alone to choose the appropriate statistical test. In view of this and given the fact that individual replicates were pooled from more samples it is recommended that the authors consult a biostatistician regarding how to best compare and analyse these data and add this new information to the manuscript. Alternatively, either the replicates for each experiment would need to be increased to an n number sufficient to determine distribution normality and to apply the corresponding test, or any information indicating statistical significances would need to be removed.

We agree that the number of replicates after pooling the samples provides a limitation for statistical analysis. In response to this comment, we carried out new experiments with three biological replicates and 4-6 technical replicates. This increased the n number to an extent that is sufficient for reliably using unpaired Student’s t-test on normally distributed data (see Figure 3).